# Humanized Mice in Dengue Research: A Comparison with Other Mouse Models

**DOI:** 10.3390/vaccines8010039

**Published:** 2020-01-22

**Authors:** Carolina Coronel-Ruiz, Hernando Gutiérrez-Barbosa, Sandra Medina-Moreno, Myriam L. Velandia-Romero, Joel V. Chua, Jaime E. Castellanos, Juan C. Zapata

**Affiliations:** 1Grupo de Virología, Vicerrectoría de Investigaciones, Universidad El Bosque, Bogotá 110121, Colombia; caritocruiz@hotmail.com (C.C.-R.); hhgutierrez49@gmail.com (H.G.-B.); mlvelandiaro@gmail.com (M.L.V.-R.); castellanosjaime@unbosque.edu.co (J.E.C.); 2Institute of Human Virology, University of Maryland School of Medicine, Baltimore, MD 21201, USA; smmoreno@ihv.umaryland.edu (S.M.-M.); JChua@ihv.umaryland.edu (J.V.C.)

**Keywords:** DENV, dengue mouse models, humanized mice, immune response, pathogenesis, vaccine, drug development

## Abstract

Dengue virus (DENV) is an arbovirus of the Flaviviridae family and is an enveloped virion containing a positive sense single-stranded RNA genome. DENV causes dengue fever (DF) which is characterized by an undifferentiated syndrome accompanied by fever, fatigue, dizziness, muscle aches, and in severe cases, patients can deteriorate and develop life-threatening vascular leakage, bleeding, and multi-organ failure. DF is the most prevalent mosquito-borne disease affecting more than 390 million people per year with a mortality rate close to 1% in the general population but especially high among children. There is no specific treatment and there is only one licensed vaccine with restricted application. Clinical and experimental evidence advocate the role of the humoral and T-cell responses in protection against DF, as well as a role in the disease pathogenesis. A lot of pro-inflammatory factors induced during the infectious process are involved in increased severity in dengue disease. The advances in DF research have been hampered by the lack of an animal model that recreates all the characteristics of this disease. Experiments in nonhuman primates (NHP) had failed to reproduce all clinical signs of DF disease and during the past decade, humanized mouse models have demonstrated several benefits in the study of viral diseases affecting humans. In DENV studies, some of these models recapitulate specific signs of disease that are useful to test drugs or vaccine candidates. However, there is still a need for a more complete model mimicking the full spectrum of DENV. This review focuses on describing the advances in this area of research.

## 1. Introduction

### 1.1. Virus

Dengue virus (DENV) is a member of the *Flavivirus* genus of the Flaviviridae family with approximately 11,000 nucleotides single-stranded RNA positive-sense genome that encodes three structural proteins (envelope or E; pre-membrane/membrane or pre-M/M; and the capsid, C) and seven nonstructural proteins (NS1, NS2a, NS2b, NS3, NS4a, NS4b, and NS5). These nonstructural proteins appear to play important roles in viral replication [1]. DENV are grouped into four serologically similar but antigenically distinct serotypes, DENV-1, DENV-2, DENV-3, and DENV-4; each serotypes is able to produce the same full spectrum of disease and could be recognized during the diagnostic tests by molecular tools or by specific antibodies raised during infection or a distinct host immune response [2].

The primary cells targeted by DENV in humans are mainly dendritic cells found in the dermis, and monocytes and macrophages recruited during infection [3] DENV receptors vary according to the cell type and this includes the dendritic cell-specific intercellular adhesion molecule-3-grabbing non-integrin (DC-SIGN) [4], cluster of differentiation 14 (CD14) [5], heat shock protein 70 (HSP70) [5], heat shock protein 90 (HSP90) [6], and glucose-regulated protein 78 (GRP78) [7]. Interestingly, each serotype interacts differentially with specific receptor molecules, demonstrating the versatility of the viral E protein to interact with a multitude of surface molecules on mosquito or human cells [8]. 

Once the virus-receptor interaction is established, the viral particle is internalized by clathrin-dependent or independent mechanisms (depending on the cell type) [9,10]. In the mature endosome, the change in pH favors the anchoring of viral E protein through its DII domain with the endosome membrane, and the viral RNA is released in the cytoplasm accompanied by protein C [11]. Thereafter, the process of translation and viral replication commences in deformed areas of the endoplasmic reticulum called viral replication organelles [12]. Concurrently, non-structural proteins promote RNA translation (NS3) and transcription/replication (NS2b/NS3 and NS5), modulate the innate immune response, and play a role in the assembly of the virion.

Immune response to DENV appears to be dependent on host susceptibility, viral factors, and baseline DENV-immunologic status. During primary dengue infection, immune response to dengue elicits an antibody response to the homologous serotype that is neutralizing, protective, and long lasting. It also elicits cross-reactive neutralizing antibodies that are initially protective but with titers that appear to wane over time (most believe around 6–12 months) to levels that are subneutralizing, potentially enhancing a pathologic outcome [13].

Epidemiologic evidence points to the fact that more severe dengue occur at higher frequency during a second infection with a different DENV serotype [14,15]. This led to the hypothesis that preformed antibodies do not neutralize the second serotype, but after recognizing the virus, direct virus-antibody complexes to Fc-receptors in monocytes and macrophages increasing the number of infected cells and viremia, enhancing the ability to cause an exacerbated disease as evidenced by higher viral load in people with severe dengue. These severe DENV cases are associated with an extensive T-cell activation and an aberrant humoral response that affects the endothelium structure and function [16]. Antibody-dependent enhancement (ADE) has been proposed as the mechanism that explains higher rates of severe disease in secondary heterologous DENV infection [15]. The concern for vaccines inducing ADE has been one of the major factors that has hindered the development of a vaccine capable of inducing a robust and balanced immunologic response against all four serotypes simultaneously [13].

To date, there is no available antiviral agent approved for the treatment of dengue. Though there are several multivalent vaccines in development, only Dengvaxia, a recombinant live attenuated tetravalent DENV vaccine with a yellow fever YF 17D backbone, has been approved by the U.S. Food and Drug Administration (FDA) for use in a specific population. Dengvaxia has been constrained to people from 9 to 16 years of age with laboratory-confirmed previous dengue and who lives in endemic areas. A limitation of the current vaccines is that the efficacy across serotypes remains uneven [17,18], and long term data points to an increased risk of hospitalizations in those individuals not previously exposed to dengue [19].

The correlates of protection for dengue are not well defined. Although, the presence of homologous neutralizing antibodies titers were initially considered to be associated with protection, recent vaccine trials found that some participants with neutralizing antibodies developed dengue [20,21].

### 1.2. Epidemiology

DF is the most prevalent mosquito-borne viral illness in the world with an estimated 390 million infections annually and 96 million symptomatic infections per year [22]. Dengue occurs in at least 128 countries in the subtropical and tropical regions of Asia, America, and Africa, with an estimated 3.9 billion people at risk of infection [23,24].

The geographic distribution of dengue correlates with the distribution of its primary vector, the urban-adapted Aedes aegypti mosquito, which circulates all year round in tropical and subtropical regions of the world. Sporadic infections occur in non-endemic areas usually diagnosed from travelers from endemic regions. In endemic areas, dengue outbreaks result in a heavy burden in terms of cost and loss of productivity for the affected families, communities, and healthcare systems [3].

The classic DF presents with self-limiting but prominent high-grade fever usually accompanied by headaches, retro-orbital pain, myalgia, arthralgia, rash, leukopenia, and thrombocytopenia. It has an incubation period between 4 to 7 days after a mosquito bite. Viremia usual occurs during the febrile period. The majority of DENV cases are asymptomatic or clinically inapparent and only recognized after serologic surveys [22]. Nevertheless, a small proportion (<5%) of patients develop potentially life-threatening severe disease known as dengue hemorrhagic fever (DHF) and dengue shock syndrome (DSS). Clinically, the abrupt development of hemorrhagic manifestations with or without hypotension (or shock) during defervescence are the hallmarks of DHF/DSS. As mentioned above, this leads to the idea that severe disease is possibly due to an immunologic damage [25,26,27,28]. Severe dengue tends to occur in hyper-endemic areas where all four serotypes circulate and is characterized by increased vascular permeability and uncontrolled viremia generally 10–100 fold higher than is seen in the common less-severe dengue fever [29,30]. 

### 1.3. Immune Response and Pathogenesis of Dengue Virus

During mosquito feeding, DENV is inoculated into the dermis and epidermis, and some virus is also injected directly into the blood-stream. In the epidermis, cells such as keratinocytes and Langerhans cells are infected, while in the dermis the fibroblasts, resident dendritic cells (DC) and mast cells are the prime targets of infection [31]. 

Once DENV enters the cells, it activates different molecular pattern recognition receptors (PRR) such as: RIG-I and MDA5 sensing cytosolic and cytoplasmic viral RNA/dsRNA [32]. Its double-stranded RNA (dsRNA) activates endosomal toll-like receptor 3 (TLR3), and its single-stranded RNA (ssRNA) activates TLR7 and TLR8 [33]. DENV also activates the complement system and TLR9/cGAS [34,35,36]. The activation of these PRRs initiates the antiviral response with a strong and early production of interferon-β (IFN-β), a weak production of interferon-γ (IFN-γ), and antimicrobial peptides (AMP) in cells such as keratinocytes [37]. In fibroblasts the infection produces IFN-β, tumor necrosis factor-α (TNF-α), interleukin-4 (IL-4), IL-1-β, chemokine (C-C motif) ligand 2 (CCL2), and chemokine (C-C motif) ligand (CCL5) [38]. On the other hand, cells such as mast cells up-regulate the secretion of CCL5 (RANTES), IL-8, IL-6, IL-10, IL-25, IL2-R, as well as endothelial growth factor (VEGF-A). The latter activates endothelial cell proliferation and is associated with increased vascular permeability. In addition, it has angiogenic and lymphogenic properties that together with the presence of viral particles in both cytoplasmic granules and granules after their release from mast cells, facilitate the systemic spread of the virus [39].

When the antiviral response is triggered, there is a massive migration of circulating monocytes to the site of infection, subsequent maturation to macrophages inducing an up-regulation of the major class II histocompatibility complex (MHC II), and together with the dendritic cells (DCs) an stimulation of antigen-presenting cell function [40]. An association between the disease severity and the decrease in the relative proportion of myeloid DCs (mDCs) as well as changes in their maturation process has been described [41]. This decrease in mDCs could be related to infection-induced apoptosis, however the mechanism of this event has not yet been fully described [42]. This event negatively impacts the antigen presentation and T lymphocyte (CD8+, or CD4+) activation by the major histocompatibility complex (MHC) encoding human leukocyte antigen (HLA).

Currently, HLA variants related to protection or susceptibility to DF have been described. In the Thai population HLA-A*0203 was described as a protective factor, while HLA-A*0207 is considered as a risk variant [43]. However, in Brazilian populations, HLA-DR*13 was associated with protection against secondary infections while HLA-B*44 was associated with disease susceptibility [44]. 

The interaction of components of both innate and adaptive immune response during DENV may differ, because of the interaction of some factors such as the host genetic characteristics, the type of infection (primary or secondary) and characteristics of the virus, which may have an essential influence on the pathogenesis of the disease [45,46,47].

The process of innate immunity activation induces the migration of some of these subpopulations to lymphoid nodules for antigen presentation to naive T lymphocytes, inducing activation, expansion, and differentiation to distinctive phenotypes of effector T cells and the production of memory T cells [48]. CD4^+^ T lymphocytes polarized to the TH0 and TH1 phenotypes producing cytokines such as IFN-γ, TNF-α, IL-2, and chemokines such as MIP-1β, while TH2 lymphocytes producing cytokines of which IL-4 and IL-10 have been the most described. In addition to contributing to the production of these cytokines, these cells enhance the response of CD8^+^ T cells and B lymphocytes for infection control [49,50,51]. Simultaneously, CD8^+^ (CD38^+^ HLA^−^ DR^+^) T lymphocytes activate and exert their cytotoxic activity mediated by IFN-γ, TNF-α, granzymes, and perforins, promoting viral clearance, and the elimination of infected cells [52,53]. 

After a primary infection, an individual may have a second exposure to a homologous or heterologous serotype of DENV, increasing the susceptibility to develop a secondary infection. This infection is characterized by a faster cellular and humoral immune response compared to the primary infection, which is characterized by the activation of different cell populations and a significant production of antibodies [50]. 

One of the hypotheses contributing to the pathogenesis of the dengue response concerns the Original Antigenic Sin [51,54,55]. This is explained by the pre-existence of memory cells that have low avidity for the new serotype of DENV and predominate in response to the new infecting serotype [56]. Because of the low effectiveness of memory cells to eliminate infected cells and the failure of viral clearance, there is a window of time of uncontrolled viral replication, contributing to the severity of the disease [51,57,58]. In addition, there are differences in T-cell responses due to peptide variants between DENV serotypes. Within these differences, some variations have been studied in the response pattern of T lymphocytes in the production of cytokines, which triggers the phenomenon called cytokine storm, which is characteristic of the pathogenesis of the disease [55,59]. 

In secondary infections, it has also been reported that the pre-existence of antibodies against proteins of the virus has a dual role, inducing protection against infection or causing damage. For example, the activity of antibodies against the NS1 protein of DENV, which in addition to forming complexes with the NS1 protein of the virus, recognize epitopes of some proteins such as fibrinogen and epitopes present in host cells such as endothelial cells, and thrombocytes [60]. These antibodies have been involved with increased endothelial cell permeability, plasma extravasation, and the development of thrombocytopenia: critical processes that occur in cases of dengue [61,62]. In addition, these antibodies facilitates the internalization of virus particles, the inhibition of the IFN-α-induced antiviral responses, the induction of IL-10 production, and favors the replication and dissemination of the virus [59,63]. This phenomenon is called ADE, and it is associated with an increase in viremia, production of high cytokine levels, and production of inflammatory mediators leading to an increased risk of developing the severe disease [15,62,64]. 

## 2. Mouse Models for Dengue Research

Pre-clinical studies on DENV, have proven to be a difficult task because of the virus specificity for some subsets of human cells and the challenge for DENV clinical isolates to replicate or cause pathology in the non-human host [65]. Non-human primate models have had limited utility because of the absence of clinical signs of DENV disease with the exception of a study in rhesus macaques using unusually high titers of inoculum (10^7^ infectious particles/macaque) [66]. Unlike humans, wild-type mice are naturally resistant against DENV, as the virus does not inhibit murine interferon (IFN) signaling [67]. Additionally, some wild type (WT) murine models show signs not seen in the human disease [68,69]. Nevertheless, some clinical signs could be individually mimicked leading to the development of different mouse strains as models for DENV replication or a specific clinical sign [70]. When wild type mice, such as C3H/He, AKR, A/J, BALB/c, B6, C57BL-6, were challenged with DENV, they would develop viremia, thrombocytopenia, as well as antiplatelet antibodies, and neuropathological disease that is in some cases associated with lethal outcomes [71,72,73,74]. DENV in human beings has not been associated with central nervous system disease, and mortality in humans is usually due to complications from vascular leakage. Moreover, variability in susceptibility to DENV among wild-type mice depends on the inoculation route and DENV dose used [69]. Although not ideal, some immunocompetent mice can be useful to test some of the pathogenesis aspects of DF/DHF. Later, in order to improve mouse models, several mutations were introduced to allow better levels of infection or to find the role of specific genes in DF pathogenesis. Furthermore, the use of immunodeficient mice engrafted with human hematopoietic cells provides a permissive environment for DENV replication in the natural host cells. The following paragraphs describe some of the common mouse models used for the study of DENV.

### 2.1. Wild Type Mouse Models

Immunocompetent mice including C3H/He, AKR, A/J, BALB/c, B6, C57BL-6 were used to test DENV through different routes resulting in viremia, thrombocytopenia, and development of anti-platelets antibodies [71,72,73,74], neurological signs, and in some models death is the final outcome, with susceptibility differences among strains depending on the inoculation route and DENV dose [69]. Although these animals develop signs of disease, they remain inadequate for modeling human DENV disease (Table 1).

### 2.2. Knockout Mouse Models for the Study of DENV

Modification of certain genetic characteristics of mice allows improvements in the efficiency of viral replication of different strains of DENV. The host activation of the IFN signaling pathway is necessary for both inflammatory and antiviral responses to curtail DENV [67]. Therefore, development of IFN-deficient mouse models appears to be the next logical step to overcome this hurdle. Some Knockout mice (KO) in which specific genes were deleted to identify and evaluate components of the innate and adaptive immune responses were generated, specifically to remove the mouse IFN responses as described below (Table 1). 

#### 2.2.1. AG129

DENV is very sensitive to IFN type I, and it is the primary reason why rodents are resistant to dengue [75]. In contrast to WT mice, A129 mice lack the type I IFN receptor (IFN-α/β) and G129 mice lack the type II IFN receptor (IFN-γ). By cross breeding the A129 with G129, AG129 strain (lacking both type I and II IFN receptors) was generated in the 1990s and became a preferred model for dengue research [76]. AG129 mice infected with a mouse-adapted DENV-2 strain died irrespective of age, while immunized mice survived virus challenge; passive transfer of the anti-DENV polyclonal antibody also promoted survival, suggesting the utility of this model for vaccine and antiviral development studies [77]. Nevertheless, the AG129 model has multiple limitations including dependence on highly mouse-adapted DENV strains that only replicate in severely immunocompromised mice; the lack of other clinical symptoms (fever or rashes) seen in humans, and the absence of thrombocytopenia and leukopenia when inoculated subcutaneously [78], although thrombocytopenia and leukopenia have been observed in intraperitoneal challenge studies [79,80].

#### 2.2.2. STAT

Mouse studies showing that individual deletion of type I and II interferon signaling, together with elimination of STAT genes underline the importance of STAT-1 and STAT-2 proteins in inhibiting viral replication [81,82,83]. Individual deletion of Stat1^−/−^ or Stat2^−/−^ genes allows some levels of DENV replication without killing the mouse. However, double mutation (in Stat1^−/−^ or Stat2^−/−^) is lethal causing early death of these mice. Those studies revealed the role of STAT-2 dependent pathway in facilitating transcription of IFN-stimulated genes against DENV in the absence of STAT-1 [81].

#### 2.2.3. C57BL/6 DKO, TKO, Irf1^−/−^, Ifnar1^−/−^, TNF^−/−^, and CCR5^−/−^

Murine models with the C57BL/6 background, have been used to study type I and II IFNs role in dengue. Several studies have shown that Irf3^−/−^ Irf 7^−/−^ double knockout (DKO) mice survive dengue despite developing high levels of viremia. While Type I IFN knockout mice Irf1^−/−^ and Ifnar1^−/−^ succumbed to DENV, triple knockout (TKO) mice survived the challenge. TKO had minimal type I IFN production but robust type II IFN-γ response and if treated with an anti-IFR antibody they die, revealing the antiviral role of IRF-1 against DENV [84]. C57BL/6 CCR5^−/−^ mice infected with mouse-adapted DENV-2 were resistant to lethal infection, underlining the importance of CCR5 in viral replication and disease [85].

The models described above do not precisely reproduce what is happening in an immunocompetent organism and in the natural environment where optimal target cells are present. Therefore, other models were developed in which human genes or their equivalents can be manipulated. 

### 2.3. Transgenic Non-Humanized Model of DENV

The development of genetically-engineered mice or transgenic murine (Tg) models for DENV infection have shed light on some of the immune mechanisms of viral pathogenesis.

#### C57BL/6J hTNF^+++^, IFNα/βR^−/−^ Tg, Tg HLA-A*02:01, and B10.Tg HLA-DR3

C57BL/6J transgenic mice overexpressing human tumor necrosis factor (TNF) and infected with DENV, developed neurological symptoms, and treatment with anti TNF-antibodies reduce mortality suggesting that intervention of this pathway may be used to treat DENV encephalitis [86]. However, when infected iv, these mice do not develop viremia and mount a specific T-cell response against a high virulent DENV strain obtained in the lab. In contrast, IFNα/βR^−/−^ Tg C57BL/6J mice develop viremia and disease, recovering and clearing the virus after 6 days [52]. Approximately, 42 T-cell epitopes present in human PBMCs has been identified [87] and by using IFNα/βR^−/−^ Tg and C57BL/6J wild type mice, of 12 epitopes derived from 6 of the 10 DENV proteins were reported [52]. In addition, C57BL/6J Tg expressing human MHC alleles such as HLA, A*0201, A*0101, A*1101, B*0701, and DRB1*0101 have been developed to study T cells in the context of viral infections [87,88,89] opening the possibility of using C57BL/6J-MHC Tg or IFNα/βR^−/−^-MHC Tg to identify T-cell epitopes relevant to protection against dengue.

### 2.4. Humanized Mouse Models of DENV

Humanized mice are immunodeficient mice transplanted with human cells or tissues, with a permissive phenotype that allows integration of various human components, for study of human pathogens or disease processes in vivo. The capacity to develop identical cell phenotypes to those observed in humans allows the study of both innate and adaptive immune responses during viral processes like DENV. The models can offer a mechanism to study pathogenesis in mice with a fully intact immune response while also providing human cells that are specifically targeted by the virus during the course of natural infection, and at the same time delivering a mechanism for interrogating viruses in the absence of viral genome adaptation or disruption of the native immune response, with applications in pathogenesis, vaccine and drug development studies (Figure 1). 

Current humanized mouse models have several limitations including the time to engraftment of human stem cells and development into human lymphocytes, the variation among species and engraftment method; the experimental window limited by the developmental time of xenogeneic graft-versus-host-disease; the heterogeneity in the level of graft acceptance; and the fact that not all components of the immune system are reconstituted. Nevertheless, these human-mouse hybrids with integrated human cells and tissues allow the replication of several aspects of clinical DENV disease and are potentially useful for drug and vaccine development. After the first description of nude mice (FOXN1 mutation) in the 1960s, different types of immunodeficient mice have been developed and used for cancer, transplant, and infectious diseases studies. The fact that they can reconstitute human hematolymphoid components after human hematopoietic stem cell (HSC) engraftment and the lack of functional natural killer (NK) cells made them a good model to study DENV. Some of these mouse strains are described in the following paragraphs and in Table 2.

#### 2.4.1. SCID

Although both mouse strains lack functional T and B cells, severe combined immunodeficiency (SCID) mice are more immunocompromised than nude mice. SCID mouse harbor a genetic autosomal recessive mutation designated Prkdc^scid^ and they accept engraftment of human cells, making them susceptible to DENV [104,105,106]. In one of these studies, SCID mice were reconstituted with human peripheral blood lymphocytes (hu-PBL) and infected with DENV, resulting in a very low efficiency of infection (only 13 to 28% of the animals had detected viral loads). A possible explanation is the low number of susceptible human cells [107].

#### 2.4.2. NOD/SCID, and NSG 

After SCID mouse development other murine strains were established with enhanced immunodeficient phenotype. Among them are NOG (NOD/Shi-scid/IL-2Rγ^−/−^), NOG/SCID, and NSG (NOD SCID gamma) mice harboring a scid mutation combined with a mutation in the IL-2 receptor. Those mice after transplantation with human HSC partially reconstitute the human immune system, permitting the replication of both mouse-adapted and clinical isolates of DENV. 

NOG/SCID mice were infected by DENV mimicking a mosquito bite transmission, developing some clinical signs of dengue fever, including fever, rash, thrombocytopenia, and viremia, showing their potential utility to test antiviral compounds [108]. 

NSG mice (backcrossed NOD/SCID mouse to IL2Rgc-KO mouse) transplanted with hematopoietic human CD34+, were used to evaluate the virulence potential of different DENV clinical isolates. After infection, DENV was found in spleen, bone marrow, and lymph nodes. In addition, these animals developed fever, reduction in platelet numbers (due to megakaryocytes and megakaryocyte progenitor’s infection) and elevated amounts of some human cytokines in serum. However, it did not resemble dengue virus severe disease (DHF/DSS) and there were no functional antiviral immune response [65,109,110]. These observations highlight the importance of the cells that participate in the innate immune responses (these animals poorly reconstitute these subsets of cell populations) when using these models. The importance of human NK cells in controlling DENV in humanized NSG mice through IFN-γ secretion has been demonstrated, reducing viral replication, thrombocytopenia and liver damage. Modeling early events of viral infection after intradermal injection by mosquito bites. These NSG mice were optimized for human NK production by injecting plasmids encoding IL-15 and Flt3L [111].

#### 2.4.3. NSG-BLT and NOD/SCID-BLT

Partial reconstitution of the human immune system in NSG, NOD/SCID, or RAG mice (described later in this paper), limit the results obtained from multiple DENV studies. In other words, because of the lack of some immune cell lineages, this system does not model viral immune responses. One way to reconstitute additional components of the human immune system with a more extensive adaptive immune response is to engraft human bone marrow, liver, and thymus (BLT) followed by HSC transplantation, resulting in an improved reconstitution of the human immune system with presence of almost all immune human cells and education of them in the thymus. Although there is a better reconstitution, these animals still lack functional humoral immune response because of the absence of class-switching and affinity maturation, limiting the efforts to understand the role of antibodies in ADE and protection in vivo. There are some transgenic options to overcome this obstacle that will be discussed later. 

NSG BLT mice infected with DENV showed higher levels of specific antibodies when compared with NSG mice, secreting DENV specific IgM antibodies with neutralizing activity in serum and IFN-γ production in response to DENV HLA-A2 restricted peptides in vitro. The authors proposed this model to test human immune responses to DENV vaccines and to explore the effect of preexisting immunity on secondary infections [112]. 

NOD/SCID-BLT mice infected with a clinical isolate (DENV-2 Col), were viremic over 10 days, developed fever, thrombocytopenia, elevated serum levels of cytokines, and induced neutralizing human IgM. However, no signs and symptoms of severe disease were observed. Treatment with an adenosine nucleoside inhibitor (NITD008), decreased viremia and there was activation of T cells with effector function ex vivo when they were re-stimulated with dendritic cells infected with DENV. However, the DENV-induced T-cell response in vivo was comparable to other humanized mouse models for DENV with different background rendering this model useful for some pathogenesis and drug development studies but not for vaccine development [113].

#### 2.4.4. NSG HLA Class I Transgenic Strains, Expressing A2 Haplotypes

Murine or primate models for DENV showed specific CD4+ and CD8+ T-cell responses against viral peptides presented under these animal’s HLA context. Several human DENV CD4+ and CD8+ T-cell epitopes have been described for both structural and non-structural proteins with predominance of non-structural for CTL recognition [114]. One of the main drawbacks of HSC-humanized mice is the absence of T-cell education in human thymus. In order to overcome this limitation, NSG transgenic mice expressing human HLA class I (HLA-A2) were developed, known as NSG-A2, allowing the evaluation of human CD4+ and CD8+ T-cell responses directed at viral peptides presented on human HLA molecules. 

After humanization and DENV infection, NSG-A2 mice showed viral replication and induction of human DENV-specific immune responses evidenced by IFN-γ, IL-2, and TNF-α secretion after stimulation with three A2-restricted dengue peptides [115].

#### 2.4.5. NSG-SGM3 and NOG-EXL Mouse Model 

Some humanize transgenic NSG mice expressing granulocyte-macrophage colony-stimulating factor (GM-CSF), and interleukin-3 (commercially known as NOG-EXL), and human stem cell factor (commercially known as NSG-SGM3) were developed in order to improve myeloid cell lineage generation [116]. The expression of these factors are expected to improve T and B-cell maturation, development of secondary lymphoid structures, and better reconstitution of myeloid antigen presenting cells (APC) and cytokine production [117]. When compared with NSG-BLT mice, NSG-SGM3-BLT have higher levels of mature naïve B cells and lower levels of immature transitional and transitional B cells, higher basal levels of human IgM and IgG in plasma, and after DENV infection, higher levels of antigen-specific IgM and IgG, suggesting an enhanced development and maturation of human B cells that can be used for studies in human antigen-specific B cell responses [117]. 

#### 2.4.6. NRG

NRG mice also known as NOD-Rag1^−/−^ IL2rγ^−/−^ double mutant mice or NOD.Rag1KO.IL2RγcKO, harbor NOD and Rag1 background. Mutations in Rag1 and IL-2 receptor, plus a mutation in the gene encoding murine signal regulatory protein α (SIRPα), render NRG mice deficient in mature B and T cells, as well as lacking NK functional cells, hemolytic complement, and robust phagocytic activity. This phenotype also supports HSC engraftment with subsequent reconstitution of human hematolymphoid system [118]. The levels of human engraftment and the lack of functional mouse NK cells made these mice a good model for the evaluation of DENV. 

#### 2.4.7. RAG2 and DRAG

Mutation in both RAG-1 and RAG-2 genes in mice also induced an immunodeficient phenotype preventing maturation of B and T lymphocytes and when combined with a mutation affecting IL-2 and IL-7 receptors, the NK cell function is also compromised. These mice are known as RAG2^−/−^γc^−/−^ or RAG-hu mice. After hematological reconstitution with human CD34+ HSC, RAG-hu mice were infected with DENV resulting in viremia, fever, and induction of human anti-DENV IgM and IgG with neutralization capacity, indicating the potential use of this model in dengue immunopathogenesis and vaccine studies [119]. 

Another transgenic immunodeficient mouse constitutively expressing human HLA-DR4 and with NOD.Rag1KO.IL2RgcKO background was developed to produce long-lived functional T and B cells capable of mounting a specific human humoral immune response. These mice, known as DRAG, showed elevated numbers of Tfh (T follicular helper) cells permitting the class switching after humanization with HLA-DR matched HSC. In addition, they secreted human cytokines, human IgM, IgG, IgA, and IgE in serum [120]. However, this model has not yet been tested in DENV infection but DRAG mice could be considered as a good model for humoral response studies against DENV.

## 3. Mouse Models for the Development of Dengue Vaccines

The easiest and cheapest way to control DENV is through vaccination. Several vaccines have been analyzed and licensed with mixed results. However, some studies demonstrated that T-cell responses are essential for controlling DENV and reduce ADE adverse effects [18,122,123]. Recent phase III clinical trials using a tetravalent dengue vaccine showed that some people having neutralizing antibodies were not protected from developing dengue. While the vaccine shows a protective response in adults, it increases the risk of severe disease in children. Concluding that, there were not clear correlates of protection, and neutralizing antibodies were not protective. 

In spite of the efforts and the investment to develop DENV vaccines, complete protection continues to be an elusive subject. One of the obstacles is the lack of an appropriate animal model to test effectivity against all four serotypes, modeling the potential immune pathogenesis induced by a heterologous secondary infection. An ideal model should allow viral replication and subsequently show development of protective immunity. Such models could provide safety information as well as help identify correlates of protection that are currently not well defined for DENV. There are some efforts to develop a DENV human challenge model to evaluate vaccine efficacy that will be the ideal system to unravel the poorly understood complexity of DENV immune response. However, because of the lack of specific therapy for DENV, there are still safety concerns for these kinds of studies in human beings. It is here where animal models could play an important role to understand those multiple factors involved in DENV immunopathogenesis and protection. In this direction, during past two decades several mouse models have been established and evaluated for drug testing, pathogenesis, immunogenicity, safety, and vaccine development studies. The following paragraphs describe some of those models used for dengue vaccine development, in particular the latest generated humanized mouse models.

### 3.1. Mouse Models to Test Safety and Immunogenicity of DENV-Vaccine Candidates

Injection of DENV clinical isolates and serial passages into suckling mouse brain resulted in virus-induced neurovirulence for the mice and attenuation for human beings [124,125]. The fact that these animals replicate virus locally, get sick, die after infection, and develop immune responses after vaccination, allowed investigators to use this route of inoculation to test DENV vaccine safety and efficacy in different strains of juvenile mice, among them Albino Swiss mice, BALB/CJ, Swiss CD1, BALB/c, and SCID mice transplanted with human liver cells [123,126,127,128,129]. However, these results cannot be directly extrapolated to humans for different reasons. First, immune presentation occurs in the context of a murine immune system. Second, the inoculation route does not occur naturally, and third encephalitis is not a common manifestation of DENV disease in humans. Therefore, researches are still looking for a more complete model involving human components. 

As earlier mentioned AG129 mice lacking IFN type I and II receptors, are susceptible to a mouse-adapted DENV inoculated intraperitoneally, and they succumb around 12 days after infection. Nevertheless, previous exposure to a tetravalent-attenuated DENV vaccine (PDK-53) or passive anti-DENV antibodies protect them from lethal challenge [77,130]. Additionally, administration of anti-DENV antibodies at subneutralizing levels increased the morbidity and viral titers [77]. Also, a cDNA clone derived from PDK-53 showed protection against homologous or heterologous virus challenge [131]. Other studies using a live-attenuated DENV vaccine or a non-propagating alphavirus replicon expressing truncated DENV envelope proteins showed induction of both cellular and humoral immune responses conferring protection against IC DENV challenge. While vaccine-induced humoral responses were either protective or enhancers of dengue disease, the cellular immune response was protective [126,132,133]. Therefore, this model is useful to test DENV-live attenuated vaccine safety, to study the role of both cellular and humoral immune responses, and even further the role of ADE antibodies after vaccination and injection of different doses of neutralizing antibodies. However, the lack of type I and II signaling in AG129 mice, must be considered before drawing any conclusion related with specific immune responses to DENV. 

### 3.2. Humanized Mice Models for the Development of Dengue Vaccines

#### 3.2.1. Hu-NSG

Hu-mice have the advantage of providing human target cells needed for DENV replication. However, each mouse model reconstitutes only a fraction of the immune system. For instance, CD34+-transplanted NSG mice, partially reconstitute the human immune system producing mainly lymphocyte populations. Although other cell populations may be present, they are underrepresented in these animals. In addition, there are two important factors to consider that can introduce variation when working with these models. One is the donor to donor variations and the second is that not all animals reconstitute human cells with the same efficiency. As previously described, mosquito bite DENV infection induced innate and adaptive immune responses in these models with production of proinflammatory cytokines and IgM antibodies (seen only by this route of infection) antibodies and as seen in hu-rag mice [119,121]. Even though they are neutralizing, there is no switching from IgM to IgG antibodies. This may be explained by the lack of HLA molecules in the mouse and poor development of follicular structures limiting the development of functional B cells. Nevertheless, these animals have clinical manifestations similar to humans such as fever, rash, thrombocytopenia, and viremia [108,109,121] that serve as read-out markers of protection or vaccine failure.

#### 3.2.2. BLT-NSG

BLT-reconstituted animals provide a higher variability of immune cells, with the presence of monocytes/macrophages, DC, and mastocytes, that are the primary targets of DENV, but they are still missing other target cells such as human endothelial cells and liver, and other components of the coagulation system involved in DF pathogenesis. However, those animal models are one step closer to what occurs in a natural DENV infection in human beings. When compared with Hu-NSG, BLT-NSG engrafted animals showed enhanced titers of neutralizing IgM and enhanced HLA-2-restricted DENV T-cell responses after peptide stimulation [112]. However, the production of DENV-specific IgG is still absent. 

#### 3.2.3. Hu-DRAG

In order to improve T and B-cell development and functionality, DRAG mice were generated. After reconstitution with HSC, DRAG mice developed human hepatocytes, liver endothelial cells, and erythrocytes. In addition, they developed specific immune responses supporting the immunoglobulin class switching, producing specific human IgG after toxoid vaccination or P. falciparum infection [120,134]. Furthermore, the high frequency of T_FH_ (CXCR5^+^PD-1^++^) and precursor-T_FH_ (CXCR5^+^PD-1^+^) cells are highly susceptible to HIV infection, making this mouse model very appealing to evaluate HIV vaccines [135]. Although, this mouse model showed superior immune responses and class switching, it has not been tested for DENV infection or DENV immunization.

#### 3.2.4. Ideal Humanized Mouse Model for DENV Pathogenesis Studies and Vaccine Development

Although there are several models that recapitulate some aspects of dengue disease, there is still a need of a model that reproduces DENV transmission, replication, and severe disease allowing investigators to identify susceptibility factors, correlates of protection, the role of sequential DENV infections with different serotypes, and proceed with drug and vaccine development. 

The ideal model will be an animal model with reconstituted human immune system (e.g., BLT) presenting antigens in the human HLA context (e.g., NSG-A2 or DRAG), constitutively expressing human cytokines necessarily for immune cell development (e.g., NSG-SGM3 or NOG-EXL), with human skin to mimic viral transmission [136,137], with human liver to model liver damage caused by DENV [138], and with endothelial cells responding to human cytokines and mimicking plasma leakage (Figure 2). Such a model does not exist yet but with the current technology it is a matter of time to have it available to study DF complexity and to develop vaccine and treatment strategies. 

## 4. Treatment

There is currently no approved antiviral therapy against dengue. Drug development has long been hindered by the lack of a suitable animal model that can closely duplicate dengue infection in humans, where potential antiviral candidates can be readily tested. As mentioned previously, mice are not the natural host of dengue because of different interferon responses that are able to resist viral replication and subsequent associated disease states. When mice are administered a high inoculum of the virus, they develop clinical syndromes that are not typically seen in human infection such as nonlethal and lethal central nervous system infection [92]. In addition, DENV have been modified to be mouse-adapted to increase infectivity in these mouse models [92,139].

The ideal platform for evaluating drug candidates for DENV should have the ability to support viral replication, at the same time mimic human clinical syndromes in a consistent and predictable manner, and preferably able to demonstrate plasma leakage-induced DHF/DSS-like signs. As stated above, Hu-mice recapitulate some of these characteristics in addition to allowing replication of DENV primary isolates with demonstrated viremia [65,109,113,119]. Frias-Staheli et al. used BLT-NOD/SCID mice infected intravenously with a low-passage DENV-2 clinical isolate and treated with either intraperitoneal or intravenous injections of an adenosine nucleoside inhibitor of DENV resulting in decrease in circulating viral RNA when administered simultaneously or 2 days post infection [113]. This proof-of-concept study helps to pave the way for the development of humanized mouse models for use in drug development. 

The goals for an antiviral platform will be the reduction or abolition of DENV viremia; reduction in intensity, and incidence of clinical symptoms such as fever and rash as well as infection-related lab abnormalities after administration of antivirals. Drug strategy includes administration of pharmacotherapeutic candidates pre-DENV challenge or post-DENV challenge at different time points, to evaluate efficacy either as a prophylactic or primary therapeutic agent, respectively. 

## 5. Conclusions

Current humanized mouse models have several limitations that will need to be addressed and improved. This includes variability in level of engraftment between batches with engraftment of human cells in various studies showing a wide range (usually between 20 to 80%). Moreover, the time needed for developing human-mouse hybrids requires a longer process of preparation and subsequent engraftment. Additionally, published experiments of humanized mouse models for DENV have failed to consistently demonstrate plasma leakage and hemorrhagic phenomenon seen in severe human DENV infection. Although RAG2 showed some clinical similarities of possible plasma leakage in about 5% of mice tested, more work still needs to be done to investigate further the potential utility of humanized mouse models as a DENV antiviral platform. Lastly, each model has to be selected based on simple questions in specific areas of the disease. However, with the rapid development of new transgenic and humanized models these questions are becoming more complex and involve more elements of the human system that can be studied in DENV infections. Humanized mouse studies can help to unveil the role of each human component in pathogenesis of DF and generating new hypothesis in dengue immune responses and pathogenesis.

## Figures and Tables

**Figure 1 vaccines-08-00039-f001:**
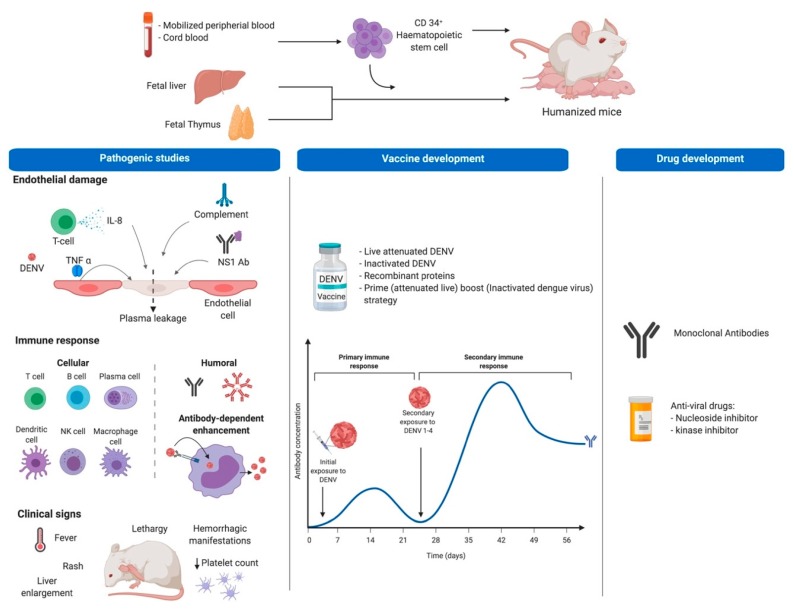
Humanization procedure and potential uses of Hu-mice for dengue studies. Top part shows tissues or organs used for obtention of CD34+ human hematopoietic stem cells (HSC) to be injected in immunodeficient mice. The bottom section shows the potential use of Hu-mice in dengue research including pathogenesis studies as well as vaccine and drug development. For pathogenic studies, researches look for development of DF clinical signs and for the role of some of the immune system components on disease pathogenesis using in vivo models, in this case Hu-mice (first panel). Additionally, Hu-mice allow testing of immunogens with potential use as vaccines protecting against all DENV serotypes (meddle panel), and since they allow viral replication, they are suitable to test compounds with antiviral activity (last panel). This figure was created under a paid subscription from Biorender.com.

**Figure 2 vaccines-08-00039-f002:**
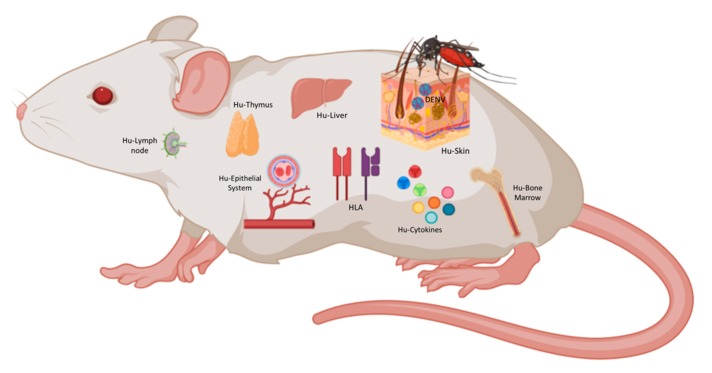
The ideal humanized mouse for DENV. As shown in this figure, an immunocompromise mouse with human HLA and human cytokines background could be transplanted with human skin, human thymus, human CD34+ HSC, and liver hepatocytes generating most of the tissues and cells affected during DENV infection. This figure was created under a paid subscription from Biorender.com.

**Table 1 vaccines-08-00039-t001:** Immunocompetent murine models or transgenic no humanized models to study dengue virus (DENV).

Mouse Model	Route	Asymptomatic Infection	Neurologic Signs	Hematological Changes	Disease Signs	Pro-Inflammatory Cytokines	Ref.
BALB/c	IP, I.C.	NR	Yes		Yes	NR	[74]
A/J	IV	NR	Yes	Transient thrombocytopenia	NR	NR	[71]
	IV	Yes	Yes	Transient ↑ hematocrit and ↓ WBC count	NR	NR	[90]
	IV	NR	Yes	Hematocrit: Slight ↑	NR	NR	[91]
	IV	NR	NR	NR	Vascular permeability: High	NR	[92]
	IV	NR	Yes	NR	Yes	IFN-γ: ↑	[93]
	IP	NR	NR	NR	Yes	IFN-γ, IL-6, IL-12, TNF-α: ↑	[94]
	IP	Yes	Yes	↑ levels of transaminases	Vascular leakage: High	IFN-γ, IL-6 and TNF-α: ↑	[77,95]
	IP	Yes	Yes	Early leukopenia	Yes	IL-1α, IL-6, IL-12p40, IL-17A, IFN-γ and TNF-α: ↑	[96]
	IP	NR	NR	Thrombocytopenia	Plasma leakage	IL-2, IL-4, and TNFα: ↑	[97]
	IP	NR	NR	Coagulopathy	Plasma leakage	IL-2, IL-4, and TNFα: ↑	
	IP	NR	NR	Thrombocytopenia	Plasma leakage	IL-1α, IL-6, IL-10, IL-12p40, IFN-γ and G-CSF: ↑	[80]
	IP	NR	Yes	Leukopenia Thrombocytopenia	Yes	IL-1α, IL-6, IL-10, IL-12p40, IFN-γ, and G-CSF: ↑	[98]
A129	IV	NR	Yes	NR	Yes	IFN-γ: ↑	[93]
	IV	NR	Yes	NR	Yes	TNF-α and IL-10: ↑	[99]
	IV	NR	NR	Thrombocytopenia and Lymphopenia: Moderate AST: Elevated	Severe liver damage Vascular leakage Middle	DENV-2: IL-6: ↑	[100]
IFNAR^−/−^	IV	NR	Yes	NR	NR	TNF-α and IL-10: ↑	[99]
	IV	NR	NR	NR	Yes	NR	[52]
	IP	NR	NR	NR	Yes	IL-6, INF-α, IP10, IFN-a, IFN-γ: ↑	[101]
Cardif^−/−^ C57BL/6	IV	Yes	NR	NR	NR	IFN-α: ↑	[102]
STAT1^−/−^	IV	NR	NR	NR	NR	IFNα: ↓ IFNβ: ↑	[81]
	IC	NR	Yes	NR	Yes	NR	[103]
STAT2^−/−^	IV	NR	NR	NR		IFNα: ↓ IFNβ: ↑	[81]
STAT1^−/−^ STAT2^−/−^and STAT1^−/−^ IFNAR^−/−^	IV	NR	NR	NR	NR	IFNα: ↓ IFNβ: ↑	

**Route of inoculation:** intravenous (IV), intraperitoneal (IP), intracardiac (IC), intraperitoneal (IP), intracranial (I.C.). **Clinical description** (presence of some of these signs). **Neurologic signs:** incoordination, loss of balance, kyphoscoliosis, and partial paralysis of posterior limbs, posture instability, ataxia, involuntary contractions, spasmodic movements, and limb paralysis. **Disease signs:** fur ruffling, weight loss, hunchback posture, abdominal distention, diarrhea-like, limited mobility, intestinal hemorrhage at 8 days after infection, erythema, increased temperature or fever. **Thrombocytopenia:** changes in platelets count comparing each mouse model with control group. **Proinflammatory cytokines:** change in cytokines levels compared with control group. Increase: (*↑*), decrease: (*↓*). **NR:** not reported.

**Table 2 vaccines-08-00039-t002:** Humanized mouse models for dengue virus.

Mouse Model	Route	Hematological Changes	Disease	Immune Response			Ref
				Pro-Inflammatory Cytokines	Cellular Response	Humoral Response	
NSG mice also known as: - NOD/SCID- NOD-^scid^ IL2rγ null - NOD-^scid^ IL2rg−/− - NOD.Cg-Prkdc^scid^IL2rg ^tm1wjl^/Sz	ID	Thrombocytopenia: High	Yes	IFN-γ, TNF-α, IL-2 and VEGF: ↑	CD4 Th1, TH2 and B cells	NR	[65]
	ID, Mosquito	Thrombocytopenia	NR	IFN-γ, IL-4, Il-10: ↑	NR	IgM, IgG, IgA	[121]
	SC	Thrombocytopenia	Yes	NR	NR	IgM, IgG, IgA	[109]
	IV	Thrombocytopenia: High hTPO; Middle Megakaryocytes: ↓	NR	NR	NR	NR	[110]
	SC, IP	NR	NR	NR	CD4+	IgM	[115]
NSG-BLT mice (HLA-A2) mice	SC	NR	Yes	IFN-γ: ↑	CD8+	IgM	[112]
NSG-BLT	IV	Thrombocytopenia: (7 or 14 dpi)	Yes	IL-R1A, VEGF, IP10: ↑ (3dpi)	CD4+, CD8+	IgM	[113]
SCID-HuH-7	IP	NR	NR	NR	NR	Neutralizing Antibodies	[106]
RAG2^−/−^ gamma c^−/−^	SC	NR	Yes	NR	NR	IgM, IgG	[119]
NSG-Tg (HLA-A2/H2-D/beta2M)1DVs/Sz mice also known as: -NOD-scid IL2rgamma null TG (HLA-A2/HuBeta2M)	SC	NR	Yes	IFNγ, TNFα, IL-2: ↑	NR	IgM, IgG, IgA	[115]

**Route:** subcutaneous (SC), intravenous (IV), intraperitoneal (IP), intradermal (ID). **Thrombocytopenia:** changes in platelets count comparing each mouse model with control group. **Disease:** fur ruffling, weight loss, hunchback posture, abdominal distention, diarrhea-like, limited mobility, intestinal hemorrhage at 8 days after infection, erythema, increased temperature or fever. **Proinflammatory cytokines:** change in cytokines levels compared with control group. Increase (*↑*) decrease (*↓*). **NR:** not reported.

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
