# Peer review of "Humanized Mice in Dengue Research: A Comparison with Other Mouse Models"

_vaccines, 2020, doi:10.3390/vaccines8010039_

Round 1

Reviewer 1 Report

Coronel-Ruiz and colleagues have provided a review regarding the integration of humanized mouse strains in the investigation of Dengue virus (DENV) infection and pathogenesis, including antiviral and vaccine development. 

This is certainly an important topic and one that has been hampered by the lack of a suitable animal model for such purposes. The authors have attempted to provide a broad overview of the current state of knowledge for mouse models in DENV infection studies. However, there were numerous concerns that were recognized in the current submission that need to be rectified prior to consideration for acceptance. These will be outlined below.

Major concerns:

The consistent use of abbreviations and proper terminology. DENV should be used following its first introduction and always used in reference to infection (not Dengue infection). There were numerous instances of this throughout the manuscript. This was also the case when discussing Dengue fever. This includes the use of both Dengue fever and Dengue disease in the abstract. Also please check ICTV for the correct use of taxonomic descriptions: https://www.ncbi.nlm.nih.gov/pmc/articles/PMC5370391/ The title. Please use "Humanized mice" in place of "Humanized mouse" Missing references. There were numerous missing references throughout the manuscript. Any section that eludes to a specific study or set of observations should be referenced. For example, lines 47, 234-236, 238, 337-343, 359-362 (should have multiple references), 417-421. Line 165: "to the new to the infected serotype" needs to be rewritten Line 181: ADE abbreviation already introduced previously Lines 203-204: this is assuming that the cells being engrafted are those that are primary targets for the virus during infection Wild-type mouse models section: These sections are somewhat concerning due to their brevity. The tables should summarize the material but it does need to be presented here. Some reference to the prior studies needs to be provided. Case in point: in section 2.2 there is quite a bit of superfluous information regarding IFN deficient mice is presented; however, no information regarding the observations in DENV infection studies is provided. This needs to be addressed. Section 2.2.1 actually does a decent job of introducing this information. Line 209: "neurological signs and in some models dead,..." needs to be rewritten for clarity Section 2.2.2: Is this information only from a single study (reference 82)? If so, the reference should be made at the end of this sentence. If not, additional references should be provided. Lines 253-254: for DENV replication? Do the authors mean DENV infection? Line 261: In regards to 42 T cell epitopes, be specific here. How many were identified? At the very least, say "approximately" if this is what the original authors stated Table 1: There's a ton of information in this table that should be better summarized in the individual sections of the review. Lines 282-285: This statement needs to be reworded. The models can provide a mechanism to study pathogenesis in mice with a fully intact immune response while also providing human cells that are specifically targeted by the virus during the course of natural infection. Thus, it's more than just allowing for the study of viral replication kinetics. Humanized mice can provide a mechanism for studying viruses in the absence of viral genome adaptation or disruption of the native immune response. Figure 1: Perhaps listing these sections (top, etc) as parts A, B, etc. would be more beneficial? Lines 309-310: Are the authors referring to the fact that SCID mice lack these lymphocytes OR that SCID mice are more immunocompromised than nude mice? The meaning is unclear in this sentence. Line 312: lymphocytes or leukocytes? PBLs traditionally refers to peripheral blood leukocytes Line 313: low efficiency as demonstrated by viremia, clinical signs, seroconversion? Please state. Line 314: Were specific potential human cell subsets provided or postulated by the authors of this citation? Lines 316-320: It would be beneficial to describe what the differences are amongst these different strains Lines 321-323: what was observed in regards to viremia or lethality? Lines 329-330: How do you mean "no functional antiviral immune response"? The authors should provide some context for this statement. Was it a lack of IFN response? No proinflammatory response? Lines 332-333: this statement requires additional context as it doesn't really fit or seem to have a particular purpose. Line 337: but why does it limit the results obtained? The authors need to ensure that they provide context for their statements. Lines 346-347: Sentence needs to be rewritten for clarity (is the first word supposed to be "secreted" or "secreting"?) Lines 350-351: which clinical isolate? This information should be provided. Also, the animals were viremic over 25 days? Replicated the virus is not an overly technical term. In regards to cytokines, was there any particular pattern noted (i.e. proinflammatory)? Lines 352-353: Which adenosine nucleoside compound? Be specific with this. Lines 371-373: the placement of these abbreviations implies that NOG-EXL is the abbreviation for interleukin-3. Perhaps the authors could say "...IL3 (strain NOG-EXL)...". Same for human stem cell factor. Section 2.4.6: So, the authors provide a good description of NRG mice here but then say very broadly that they are good for evaluating DENV infection. But why? What has been reported in the literature that suggests this? More specifics please. Lines 397-402: This paragraph should be minimized considering that there is no information about the strain in regards to DENV infections. There are two section 3.2.1s. Also, the authors should discuss potential donor-to-donor variations when working with humanized mice.

Author Response

We would like to acknowledge the time you took to review our manuscript and your valuable comments to improve it. Below you can find our answers to each of your comments.

Comments and Suggestions for Authors

Coronel-Ruiz and colleagues have provided a review regarding the integration of humanized mouse strains in the investigation of Dengue virus (DENV) infection and pathogenesis, including antiviral and vaccine development. 

This is certainly an important topic and one that has been hampered by the lack of a suitable animal model for such purposes. The authors have attempted to provide a broad overview of the current state of knowledge for mouse models in DENV infection studies. However, there were numerous concerns that were recognized in the current submission that need to be rectified prior to consideration for acceptance. These will be outlined below.

Major concerns:

The consistent use of abbreviations and proper terminology. DENV should be used following its first introduction and always used in reference to infection (not Dengue infection). There were numerous instances of this throughout the manuscript. This was also the case when discussing Dengue fever. This includes the use of both Dengue fever and Dengue disease in the abstract.

Also please check ICTV for the correct use of taxonomic descriptions: https://www.ncbi.nlm.nih.gov/pmc/articles/PMC5370391/

R/ Those issues were corrected throughout the whole manuscript.

The title. Please use "Humanized mice" in place of "Humanized mouse" Missing references.

R/ The title was modified as suggested by the reviewer.

There were numerous missing references throughout the manuscript. Any section that eludes to a specific study or set of observations should be referenced. For example: lines 47, 234-236, 238, 337-343, 359-362, 417-421

lines 47:

R/ References added

lines 234-236:

R/ Clarification added

line 238:

R/Reference added

lines 337-343:

R/ Three references added

lines359-362:

R/ This is the only study we found,

lines 417-421:

R/ This is a statement made by the authors, there is not reference 

Line 181: "to the new to the infected serotype" needs to be rewritten

R/ It was replaced by “response to the new infecting serotype”

Line 181: ADE abbreviation already introduced previously

R/ It was deleted.

Lines 203-204: this is assuming that the cells being engrafted are those that are primary targets for the virus during infection 

R/ We agree with the reviewer. We added human “hematopoietic” cells to be more precise.

Wild-type mouse models section: These sections are somewhat concerning due to their brevity. The tables should summarize the material but it does need to be presented here. Some reference to the prior studies needs to be provided.

R/ We agree with the reviewer that this section is short and it is due the focus of this especial issue called: Humanized Mice in Vaccinology: Opportunities and Challenges.

Case in point: in section 2.2 there is quite a bit of superfluous information regarding IFN deficient mice is presented; however, no information regarding the observations in DENV infection studies is provided. This needs to be addressed. Section 2.2.1 actually does a decent job of introducing this information. 

R/This is an introductory comment to state the importance of IFN in the development of these animal models for DENV studies. As you mention the topic is developed in the next two paragraphs

Line 209: "neurological signs and in some models dead,..." needs to be rewritten for clarity.

R/ For clarification proposes this sentence was rewritten as follows: “dead is the final outcome”.

Section 2.2.2: Is this information only from a single study (reference 82)? If so, the reference should be made at the end of this sentence. If not, additional references should be provided. 

R/ Two references were added describing STAT1 deficient mice.

Lines 253-254: for DENV replication? Do the authors mean DENV infection? 

R/ Yes. This sentence was changed accordingly.

Line 261: In regards to 42 T cell epitopes, be specific here. How many were identified? At the very least, say "approximately" if this is what the original authors stated.

R/ Thank you for pointing this out. We modified this paragraph and added refences in order to be more precise.

“C57BL/6J transgenic mice overexpressing human tumor necrosis factor (TNF) and infected with DENV, developed neurological symptoms, and treatment with anti TNF-antibodies reduce mortality suggesting that intervention of this pathway may be used to treat DENV encephalitis (85). However, when infected iv, these mice do not develop viremia and mount a specific T cell response against a high virulent DENV strain obtained in the lab. In contrast, IFNα/βR−/− Tg C57BL/6J mice develop viremia and disease, recovering and clearing the virus after 6 days (53). Approximately, 42 T cell epitopes present in human PBMCs has been identified (Krishnakumar et al) and by using IFNα/βR−/− Tg and C57BL/6J wild type mice, of 12 epitopes derived from 6 of the 10 DENV proteins were reported (53). In addition, C57BL/6J Tg expressing human MHC alleles such as HLA, A*0201, A*0101, A*1101, B*0701, and DRB1*0101 have been developed to study T cells in the context of viral infections (Krishnakumar et al, Wang et al, Dimario et al), opening the possibility of using C57BL/6J-MHC Tg or IFNα/βR−/−-MHC Tg to identify T cell epitopes relevant to protection against Dengue”.

Table 1: There's a ton of information in this table that should be better summarized in the individual sections of the review. 

R/ We agree with the reviewer. However, to make the manuscript simple to read, we decide to avoid detailed information about each study.

Lines 282-285: This statement needs to be reworded. The models can provide a mechanism to study pathogenesis in mice with a fully intact immune response while also providing human cells that are specifically targeted by the virus during the course of natural infection. Thus, it's more than just allowing for the study of viral replication kinetics. Humanized mice can provide a mechanism for studying viruses in the absence of viral genome adaptation or disruption of the native immune response. Figure 1: Perhaps listing these sections (top, etc) as parts A, B, etc. would be more beneficial? 

R/ As suggested by the reviewer, this paragraph was modified as follows:

“The models can offer a mechanism to study pathogenesis in mice with a fully intact immune response while also providing human cells that are specifically targeted by the virus during the course of natural infection, and at the same time delivering a mechanism for interrogating viruses in the absence of viral genome adaptation or disruption of the native immune response, with applications in pathogenesis, vaccine and drug development studies”.

Lines 309-310: Are the authors referring to the fact that SCID mice lack these lymphocytes OR that SCID mice are more immunocompromised than nude mice? The meaning is unclear in this sentence. 

R/ As suggested by the reviewer, this sentence was modified as follows:

“Although both mouse strains lack functional T and B cells, severe combined immunodeficiency (SCID) mice are more immunocompromised than nude mice”.

Line 312: lymphocytes or leukocytes? PBLs traditionally refers to peripheral blood leukocytes 

R/ In this context, after PBMC injection into immunocompromised mice, only lymphocyte cells survive. Therefore, when referring to hu-PBL, the literature refers to mice reconstituted with human lymphocytes.

Line 313: low efficiency as demonstrated by viremia, clinical signs, seroconversion? Please state. 

R/ As stated in parenthesis “low efficiency” is demonstrated by the fact that only 5 animals out of 19 had detectable viremia. We included the percentage of efficiency of infection instead.

Line 314: Were specific potential human cell subsets provided or postulated by the authors of this citation? 

R/ Yes. The authors tested injections to mice with infected monocytes and failed in establishing a DENV infection. Therefore, they concluded that “The low level of virus in the blood of the hu-PBL-SCID mice probably reflects the low proportion of human cells found circulating by FACS analysis.

Lines 316-320: It would be beneficial to describe what the differences are amongst these different strains 

R/ The following two sentences were introduced in the text to explain those mutations present in those mouse strains.

Section 2.4.1:“SCID mouse harbor a genetic autosomal recessive mutation designated Prkdc scid and they accept engraftment of human cells, making them susceptible to DENV.”

Section 2.4.2: “Among them are NOG (NOD/Shi-scid/IL-2Rγ-/-), NOG/SCID and NSG (NOD SCID gamma) mice harboring a scid mutation combined with a mutation in the IL-2 receptor)”

Lines 321-323: what was observed in regards to viremia or lethality?

R/ Animals showed viremia (it was included in the text of this manuscript) and the author did not reported lethality.

Lines 329-330: How do you mean "no functional antiviral immune response"? The authors should provide some context for this statement. Was it a lack of IFN response? No proinflammatory response? 

R/ It means not protective humoral or cellular immune response against dengue.

Lines 332-333: this statement requires additional context as it doesn't really fit or seem to have a particular purpose. 

R/ To further clarify this point, we added the following sentence: “These observations highlights the importance of the cells that participate in the innate immune responses (these animals poorly reconstitute these subsets of cell populations) when using these models”.

Line 337: but why does it limit the results obtained? The authors need to ensure that they provide context for their statements. 

R/ The following sentence was added to clarify this point. “In other words, due to the lack of some immune cell lineages, this system does not model viral immune responses”.

Lines 346-347: Sentence needs to be rewritten for clarity (is the first word supposed to be "secreted" or "secreting"?) 

R/ It is secreting. Punctuation was corrected.

Lines 350-351: which clinical isolate? This information should be provided. Also, the animals were viremic over 25 days? Replicated the virus is not an overly technical term. In regards to cytokines, was there any particular pattern noted (i.e. proinflammatory)? 

R/ DENV-2 Col. There is a contradiction in this reference. Figure 2A shows viral detection up to 25 days. However, in the text they describe that no viremia was detected after 10 days of infection. Therefore, we will stick with 10 days. We replace “replicated the virus” by “were viremic”.

There was a broad spectrum in the induction of cytokines and chemokines but only three cytokines (IFN-α2, IL-10, and IP-10) showed a statistically significant correlation between they secretion and viral loads.

Lines 352-353: Which adenosine nucleoside compound? Be specific with this. 

R/ The name of the compound was included in the text.

Lines 371-373: the placement of these abbreviations implies that NOG-EXL is the abbreviation for interleukin-3. Perhaps the authors could say "...IL3 (strain NOG-EXL)...". Same for human stem cell factor.Section 2.4.6: So, the authors provide a good description of NRG mice here but then say very broadly that they are good for evaluating DENV infection. But why? What has been reported in the literature that suggests this? More specifics please. 

R/ The following clarification was added to the text: “Commercially known as”

Lines 397-402: This paragraph should be minimized considering that there is no information about the strain in regards to DENV infections. 

R/ Although there is not current information about the use of DRAG mice regarding DENV infection, we consider that this strain has the potential utility in studies involving humoral immune response against DENV so we suggest the possible use of this strain in DENV studies

There are two section 3.2.1s. Also, the authors should discuss potential donor-to-donor variations when working with humanized mice. 

R/ The following information was added to the text: “In addition, there are two important factors to consider that can introduce variation when working with these models. Ones is the donor to donor variations and the second is that not all animals reconstitute human cells with the same efficiency”.

Reviewer 2 Report

This paper examines the current progress in Dengue research focusing on a comparison between different mouse models.

The work is very well written and it's a pleasure to read it.

In my opinion the authors have made a great effort and I suggest its publication in this journal.

Just very minor suggestions.

I suggest the explanation of all acronyms (i.e. line 49: CD14, HSP, GRP; line 125: CCL; line 126: RANTES; line 152: MIP) just to be as clear as possible for all readers

line 15: maybe there is a repeat of "single"

line 81 dengvaxia instead of dengavaxia

In the tables, I would suggest to use up and down arrows instead of increased/decreased in the pro-inflammatory cytokines section. Maybe it could make the presentation more elegant 

Author Response

We would like to acknowledge the time you took to review our manuscript and your valuable comments to improve it. Below you can find our answers to each of your comments.

Comments and Suggestions for Authors

This paper examines the current progress in Dengue research focusing on a comparison between different mouse models.

The work is very well written and it's a pleasure to read it.

In my opinion the authors have made a great effort and I suggest its publication in this journal.

Just very minor suggestions.

I suggest the explanation of all acronyms (i.e. line 49: CD14, HSP, GRP; line 125: CCL; line 126: RANTES; line 152: MIP) just to be as clear as possible for all readers

R/ All acronyms were explained.

line 15: maybe there is a repeat of "single"

R/ on “single” was deleted.

line 81 dengvaxia instead of dengavaxia

R/ Dengavaxia was replaced by dengvaxia.

In the tables, I would suggest to use up and down arrows instead of increased/decreased in the pro-inflammatory cytokines section. Maybe it could make the presentation more elegant 

R/ Increased/Decreased words were changed by arrows. Increased was replaced by ↑ ­, and decreased was replaced by ↓.

Reviewer 3 Report

The article have been written in an excellent manner covering all relevant literature related to this topic. 

I could only find few errors in the formatting of references and would request the authors to recheck all references for uniformity. For example the name of journal in ref. 2 is typed wrong. Also the reference 13 has a different pattern of journal name (j Vs J in other references).

Author Response

We would like to acknowledge the time you took to review our manuscript and your valuable comments to improve it. Below you can find our answers to each of your comments.

Comments and Suggestions for Authors

The article have been written in an excellent manner covering all relevant literature related to this topic. 

I could only find few errors in the formatting of references and would request the authors to recheck all references for uniformity. For example the name of journal in ref. 2 is typed wrong. Also the reference 13 has a different pattern of journal name (j Vs J in other references).

R/ All references were updated and corrected.

Round 2

Reviewer 1 Report

The authors have done a commendable job to address all of the concerns and comments raised during the initial review. No additional concerns remain.